# Comparing the Performance of Radiation Oncologists versus a Deep Learning Dose Predictor to Estimate Dosimetric Impact of Segmentation Variations for Radiotherapy

**Amith Kamath**[*1]                                     amith.kamath@unibe.ch
**Zahira Mercado**[*1]                    zahira.mercadoaufdermaur@unibe.ch
**Robert Poel** [2]                                        robert.poel@insel.ch
**Jonas Willmann** [3]                                  jonas.willmann@usz.ch
**Ekin Ermis** [2]                                          ekin.ermis@insel.ch
**Elena Riggenbach** [2]                        elena.riggenbach@insel.ch
**Nicolaus Andratschke** [3]              nicolaus.andratschke@usz.ch
**Mauricio Reyes** [1,2]                              mauricio.reyes@unibe.ch

[1] *ARTORG Center for Biomedical Engineering Research, University of Bern,*

[2] *Department of Radiation Oncology, University Hospital Bern, University of Bern*

[3] *Department of Radiation Oncology, University Hospital Zurich, University of Zurich*

**Editors:** Accepted for publication at MIDL 2024

## Abstract

Current evaluation methods for quality control of manual/automated tumor and organs-at-risk segmentation for radiotherapy are driven mostly by geometric correctness. It is however known that geometry-driven segmentation quality metrics cannot characterize potentially detrimental dosimetric effects of sub-optimal tumor segmentation. In this work, we build on prior studies proposing deep learning-based dose prediction models to extend its use for the task of contour quality evaluation of brain tumor treatment planning. Using a test set of 54 contour variants and their corresponding dose plans, we show that our model can be used to dosimetrically assess the quality of contours and can outperform clinical expert radiation oncologists while estimating sub-optimal situations. We compare results against three such experts and demonstrate improved accuracy in addition to time savings. Our code is available at https://github.com/ubern-mia/radonc-vs-dldp.

**Keywords:** Image segmentation, Radiotherapy, Dosimetric clinical evaluation.

## 1. Introduction

Glioblastoma, accounting for about 45% of brain tumors (McFaline-Figueroa and Lee, 2018), is an aggressive malignant tumor treated with surgery, radiotherapy (RT), and chemotherapy (Stupp et al., 2005). RT aims to target the tumor while minimizing dose to healthy organs-at-risk (OAR). The planning involves a trade-off between tumor control and tissue toxicity (Scaringi et al., 2018). A critical step is the segmentation of structures, which is time-consuming when done manually, can take up to seven hours per patient in the head and neck anatomy (Das et al., 2009).

With advancements in deep learning-based auto-segmentation, the role of radiation oncologists is shifting from manually drawing to monitoring and correcting these automated

---

[*] Contributed equally

segmentations (Claessens et al., 2022). Quality checks are hence crucial since it has been reported that incorrect tumor volumes cause 25% of non-compliant treatment plans, leading to untreated tumors or harmful radiotherapy doses (Peters et al., 2010). While geometric metrics like Dice score coefficient (DSC) and Hausdorff distance are currently the de-facto metrics to evaluate segmentation quality, it has been reported that they do not correlate with dosimetric effects of contouring errors (Poel et al., 2021; Kofler et al.). In RT, it has been postulated that auto-segmentation methods must be evaluated using a diverse range of performance metrics, including impact on delivered dose (Harrison et al., 2022), which ultimately impacts clinical outcome.

**Related work:** The clinical RT community has developed standards for target contouring (Niyazi et al., 2023), which mainly includes geometrical and anatomical considerations. Dosimetry-based considerations require dose plan calculations, which are time-consuming and necessitate iterations between the radiation oncologist and dosimetrists or medical physicists. Hence, due to its time-consuming nature, dosimetric assessment of segmentation quality has not been conventionally employed in the clinics. Nonetheless, as recently pointed out by (Claessens et al., 2022), dosimetry considerations are urgently needed in the quality control process of tumor and organs-at-risk segmentations.

Previously proposed approaches to evaluate the quality of automated segmentations include methods that predict segmentation metrics, such as DSC, (Valindria et al., 2017) or use Graph Neural Networks to identify segmentation errors in radiotherapy (Henderson et al., 2022). These approaches assume that these geometric metrics reflect the quality of dosimetry, which is not the case (Poel et al., 2021; Kofler et al.). Furthermore, models predicting DSC perform poorly with low-quality segmentations (the main target of such QC system) due to a lack of representative training data for this performance range.

Other approaches have explored the use of uncertainty estimation in OAR segmentation in head and neck cancer (Cubero et al., 2023), under the premise that high uncertainty is linked with potentially low-quality segmentations. However, geometric variability and uncertainty estimates may not imply dosimetric effects (e.g., high uncertainty of a contour located in a non-dosimetrically relevant area), and uncertainties based on imaging information alone may not sufficiently guide quality assurance.

In line with dosimetry-focused quality assurance, a deep learning-based dose prediction model is utilized in (Roberfroid et al., 2024) to guide radiation oncologists on which volume slices require manual adjustments. This segmentation editing tool demonstrates potential for time efficiency while maintaining dosimetric equivalence with distribution maps produced without its assistance. In (Kamath et al., 2023b) we introduced a method that uses a deep learning-based dose predictor to assess the impact of local segmentation changes on dosimetric outcomes. However, this work focused on organ-at-risk segmentation, and not on tumor lesions, which is clinically more important due to the higher complexity of this segmentation task.

**Hypothesis and Contributions:** Beyond the state of the art, we postulate in this study that a deep learning based dose predictor can be employed within a quality control framework to detect dosimetrically worse segmentations, with levels of performance superior to human experts. We substantiate this by comparing the performance of our deep learning-based quality assurance method with that of three expert radiation oncologists, using a test dataset comprising 54 segmentation variations from brain tumor patients, and reference

dose plans produced by a clinically validated treatment planning system (Varian Medical Systems Inc., Palo Alto, USA).

To the best of our knowledge, this is the first study comparing the levels of dosimetric awareness on contour modifications between human experts and a deep-learning dose predictor model.

## 2. Methods

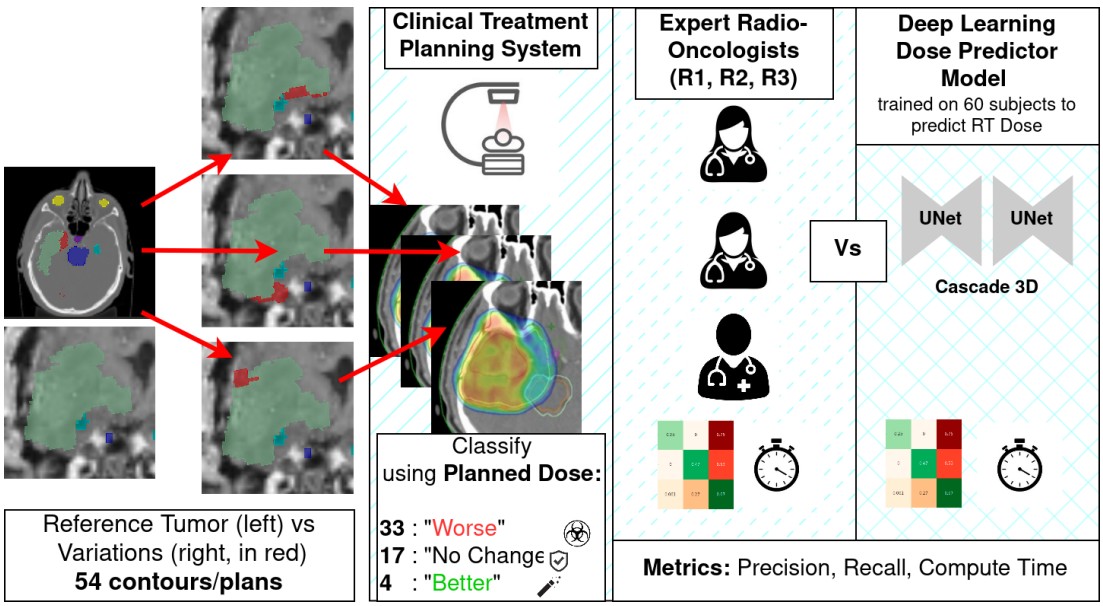

Figure 1: Is a deep learning dose prediction model able to ascertain dosimetric impact of tumor target volume contour changes when compared to radiation oncologists? An experimental study is run with 54 contour variations which are individually re-planned to generate three categories of results: "Worse", "No Change" and "Better".

Our study design, depicted in Figure 1, involves a set of reference tumor segmentations (ground-truth) and corresponding expert-derived contour variations (four per reference segmentation). For each reference and contour variation (n=54 pairs), dose plans are computed. Each contour is classified as *"Worse"*, *"No change"*, or *"Better"* based on dosimetric variations relative to the reference segmentation. This classification scheme is used to evaluate the ability of three experienced radiation oncologists and the proposed deep-learning dose predictor model to accurately classify each contour variation.

We report classification metrics and time taken to perform the task.

## 2.1. Data

Our data set includes imaging and segmentation data from 100 patients diagnosed with glioblastoma. This includes CT and Magnetic Resonance Image (MRI) T1 contrast-enhanced images, and binary segmentation masks of 13 OARs as well as the tumor target volume. The OARs include Brainstem, optic chiasm, cochlea (left and right), eye (left and right), hippocampus (left and right), lacrimal gland (left and right), optic nerve (left and right), and the pituitary gland. Each of these subjects also has a reference dose plan, calculated using a standardized clinical protocol with Eclipse (Varian Medical Systems Inc., Palo Alto, USA). This reference is a double arc co-planar volumetric modulated arc therapy (VMAT) plan with 6 mega volt flattening filter free beams, optimized (Varian photon optimizer version 15.6.05) to deliver 30 times 2 Gray while maximally sparing the OARs. The dose is calculated with the AAA algorithm (Van Esch et al., 2006), normalized so that 100% of the prescribed dose covers 50% of the target volume. All the volumes are resampled to an isometric 2x2x2 millimeter grid of size $128^3$ voxels using PyRaDiSe (Rüfenacht et al., 2023) and converted to NIfTI files to use for training and evaluation.

## 2.2. Experimental Setup

We train a cascaded 3D U-Net dose prediction model (Liu et al., 2021), which has been previously evaluated to show a mean prediction error of 1.38 Gray (Poel et al., 2023; Kamath et al., 2023a) on a subset of n=60 cases (from the original 100 cases). The inputs are the CT volume, the OAR and tumor binary segmentation masks (14 volumes), and the output is the dose prediction. Ten cases are used as validation, and 14 are used as the test set for this study (Implementation details below). We save the rest of the 16 cases for future evaluations.

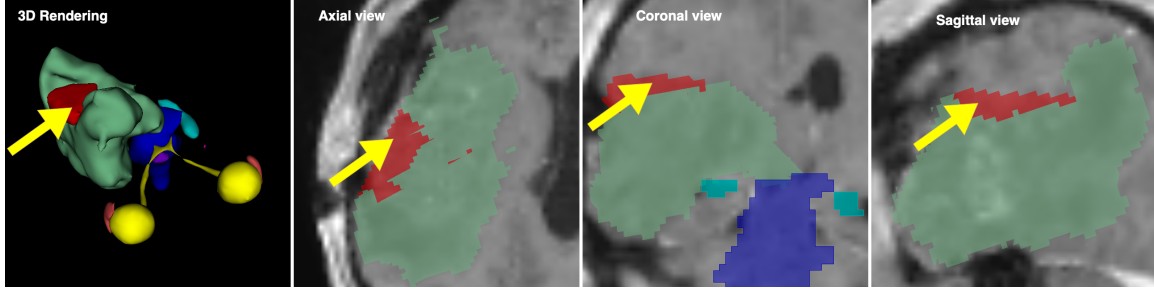

Figure 2: Example showing a tumor target volume contour change that is "Worse" - negatively impactful. The green overlay is the reference contour, red overlay indicates change, marked with a yellow arrow. OAR contours are shown for anatomic reference.

**Contour modifications and replanning:** For 14 test subjects, between three to four variations to the tumor target volume are made independently by an expert using the same rationale as in (Poel et al., 2021). The reference segmentation follows current delineation standards. The dose is then re-planned with the same settings as earlier to construct a

ground truth dose plan for each of the variations. This leads to 54 test scenarios, which are categorized into "Worse", "No Change" and "Better" from a dosimetry perspective. Figure. 2 shows how a variation (in red) looks compared to the reference (in green).

**Ground-truth dosimetric categorization of contour variations:** We define three categories of dosimetric impact based on a 10% change on the maximum dose recorded within each of the 13 OARs with respect to each reference contour as computed using the commercial treatment planning system. A contour variation is considered as dosimetrically impactful if at least one OAR crosses this threshold. We define three categories: "Worse", "No Change" or "Better" - indicating if such a change to the tumor target volume leads to higher OAR dose, no change, or lower OAR dose as compared to the reference segmentation, respectively. On the set of 54 cases, this definition leads to 33 Worse, 17 No Change, and 4 Better scenarios based on the ground truth reference dose plans.

**Deep learning based dosimetric categorization of contour variations:** To automate the categorization of contours, we use the trained dose prediction model on the reference contour and each analyzed contour variation, yielding dose maps $D_{ref}$ and $D_{cv}$, respectively. Each contour variation is then classified following algorithm 1, which follows the class definitions presented above for ground-truth generation.

---

**Algorithm 1:** Contour Variation Quality Classification - $Q(cv)$

1: **for** each $OAR$ **do**
2:     **if** $max(D_{cv}) > max(D_{ref})(1 + \alpha T)$ **then**
3:         increment counter for Worse
4:     **else**
5:         **if** $max(D_{cv}) < max(D_{ref})(1 - \alpha T)$ **then**
6:             increment counter for Better
7:         **end if**
8:     **end if**
9: **end for**
10: **if** counter for Worse $\geq$ nOAR **then**
11:     $Q(cv) =$ Worse
12: **else if** counter for Better $\geq$ nOAR **then**
13:     $Q(cv) =$ Better
14: **else**
15:     $Q(cv) =$ No Change
16: **end if**

---

Algorithm 1 examines each Organ-at-Risk (OAR) to identify those exceeding a specified percentage dose change, determined by threshold $T$. If the count of such OARs surpasses the hyperparameter $nOAR$, the case is deemed "Worse". In the absence of dose violations, the algorithm investigates potential dose improvements (line 5). If none are found, the contour variation is labeled as "No-Impact". The hyperparameter $\alpha$ serves as a calibration parameter, akin to the ROC-AUC threshold in classification models. The hyperparameter $nOAR$ sets the model's overall sensitivity to dose violations.

**Implementation details:** Each of the two U-Nets in the cascade (Liu et al., 2021) has a depth of five sets of convolution layers in the encoder and decoder with $16, 32, 64, 128, 256$

channels in the first level and twice this number in the second. The model input is a normalized CT volume and binary segmentation masks for each of the 13 OARs and target volume, and predicts a continuous-valued dose volume (up-scaled from [0, 1] to 0 to 70 Gray) of the same dimension as the input. The loss is computed as a weighted sum of L1 losses between outputs of the first and second U-Nets versus the reference dose: $Loss = 0.5 * L1(reference, A) + L1(reference, B)$, where $A$ and $B$ are the outputs of the first and second U-Nets respectively, $reference$ is the reference dose and $L1$ refers to the L1 loss. The weights are randomly initialized using the 'He' method. Training runs for 80000 iterations and the model with the best validation dose score is saved. The training batch size is set to 2 and the learning rate to $1e-3$ with a weight decay of $1e-4$. Data augmentation is done with random flipping and random rotation along the z-axis (in the axial plane). T in Algorithm. 1 is set to 0.1. All experiments are run with PyTorch 1.12 on an NVIDIA RTX A5000 GPU, and each training run takes approximately 24 hours.

**Comparing against human expert baselines:** For each test case, three expert radiation oncologists evaluated contour variations and were asked to classify them using the defined dosimetric categorization. We use 3D Slicer (version 5.6.0) for this evaluation and show all three slice planes (see Figure. 6 in the Appendix). We also include a 3D rendering of the geometric relationship between the OARs and the tumor target volume, highlighting where the contour change is made. We time their responses, and show all the variations for each subject simultaneously so that they can make visual comparisons against the reference for each contour variation.

As classification metrics, we report precision and recall, and the confusion matrices. Average time to evaluate each variant is also presented to compare performance.

## 3. Results

Table. 1 shows the weighted average (across the three categories) precision and recall as well as the average time taken to evaluate each of the 54 variants by the three radiation oncologists (in rows marked R1, R2 and R3) as compared to the deep learning dose prediction model in the last row. On both precision and recall, the model outperforms all three experts. Notably, we underline the high inter-rater variability in performance among the three experts. Expert R3, being the most meticulous and expert rater, used significantly more time than other experts. While the proposed model tends to classify more "No-Impact" contours as "Worse", we view this as a beneficial trade-off. In practice, it would lead to additional checks, which is preferable to potentially overlooking increased toxicity to the patient. The time taken by the model is dominated by two inference runs through the reference as well as the variant contours. Additionally, the range of time taken varies broadly between experts, from 19 to 138 seconds per variation, while the deep learning predictor always takes the same quantum of time irrespective of the difficulty in geometry.

Figure. 3 shows the confusion matrices (normalized by true category) for the three expert radiation oncologists (R1, R2 and R3) and the dose predictor model (right-most panel marked "Prediction"). Darker green on the diagonal is better, while darker red on the off-diagonal is not. The model outperforms all three radiation oncologists in the "Worse" category. None of the experts mark any variant as "Better".

|  | Precision | Recall | Time Taken (per variant) |
|---|---|---|---|
| **Radiation Oncologist #1** | 0.41 | 0.35 | 48 [19 - 64]s |
| **Radiation Oncologist #2** | 0.48 | 0.46 | 50 [28 - 100]s |
| **Radiation Oncologist #3** | 0.55 | **0.57** | 71 [29 - 138]s |
| **Deep Learning Dose Predictor** | **0.57** | **0.57** | 30 s |

Table 1: Precision and recall (weighted average) for each of the three expert radiation oncologists compared with model predictions. Average (max - min) time taken per variant evaluated is indicated in the last column in seconds.

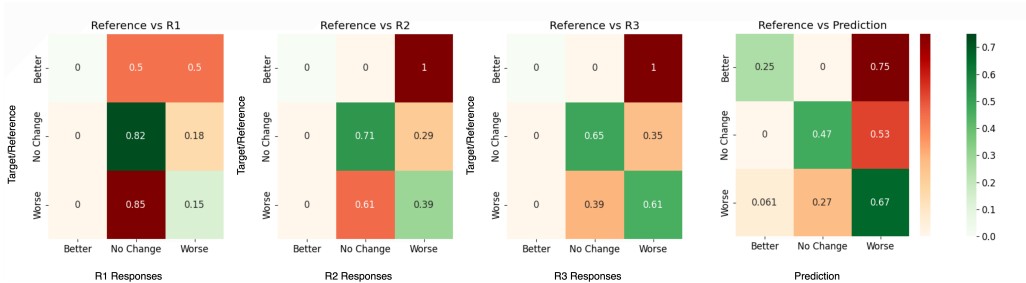

Figure 3: Confusion matrices for the classifier using the dose predictor model versus the performance of three expert radiation oncologists. Sensitive predictions imply more entries in the upper triangular region, leading to further manual checks, while still saving clinician time for correctly classified variations (on the diagonal).

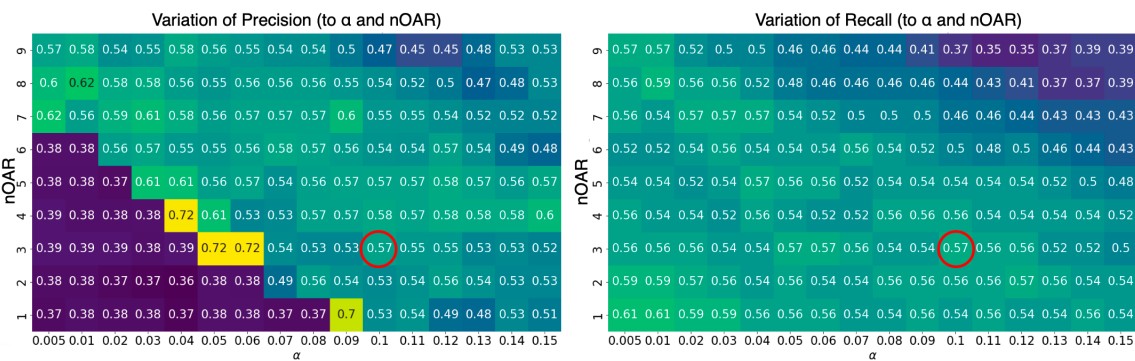

Figure 4: Performance of dose predictor model on variation of $\alpha$ and number of OARs crossing the threshold based on precision and recall. We prefer models with reasonable precision and higher recall - as we want the classification to be more sensitive in catching "Worse" plans as opposed to missing out on those that may have "No change". Red circles indicate values chosen for comparing with experts.

Figure. 4 shows the sensitivity of (weighted average) precision and recall to the hyperparameters $\alpha$ and $nOAR$ used to classify dosimetric impact. $\alpha$ (horizontal axis) ranges from 0.005 to 0.15 in each of the two heat maps. Specifically, $\alpha = 0.1$ means that the percentage change threshold for the model is 0.1 times that used for the reference (in this case, 1%). The vertical axis is $nOAR$, where smaller values make the model more sensitive and strict, and larger values increase model robustness while trading off sensitivity. Both the precision and recall metrics show a reasonably smooth variation, except that the precision values drop significantly for small $\alpha$ and $nOAR$. As good trade-off we chose $\alpha = 0.1$ and $nOAR = 3$ for the results presented in Figure. 3 and Table. 1.

## 4. Discussion and Conclusion

Radiation oncologists indicate that their mental model emphasizes proximity to OARs (closer and in the line of sight between the tumor target and OARs are more impactful) and the size of the variation (larger causes higher residual dose to OARs).

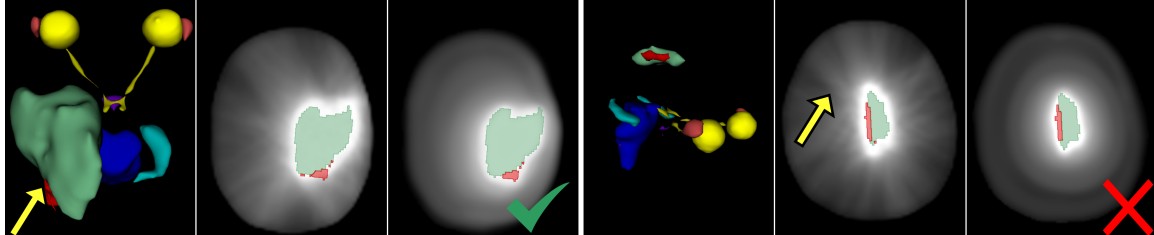

Reference: Worse; R1/R2/R3: No Change; Prediction: Worse      Reference: No Change; R1/R2/R3: No Change; Prediction: Worse

Figure 5: Two exemplar situations. Each set is shown as a 3D render, reference dose plan (axial) and predicted dose (axial). Left half: all three experts mark "No Change" due to its posterior nature (yellow arrow) away from the OARs, while the model predicts correctly. Right half: experts mark correctly as "No Change" while the model incorrectly flagged as "Worse". The yellow arrow shows the beam artifacts in the reference dose plan which are not replicated by our model.

Figure. 5 demonstrates two exemplar situations that we use to showcase the strengths as well as weaknesses of our proposed idea. The left half shows one such condition where the model correctly classifies a condition that conventionally would be considered to be not impactful. Conversely, the right half shows a situation where the model overestimates the severity. This can be attributed to the predicted dose being a smooth proxy to the actual beam structure. Recent advances (Teng et al., 2024) aim to account for this effect.

We present a novel dosimetry-driven quality control framework, where our dose predictor model outperforms human experts, indicates a promising baseline on which to build on improvements. This work demonstrates human clinician baseline, upon which we plan to work on the next set of evaluations where radiation oncologists are shown assistive maps like (Kamath et al., 2023b) to measure if their performance improves both in time and accuracy.

## Acknowledgments

We acknowledge funding by Swiss Cancer Research (KFS-5127-08-2020). We report no financial relationship or conflicts of interest.

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

## Appendix A.  User Interface for Expert Evaluations

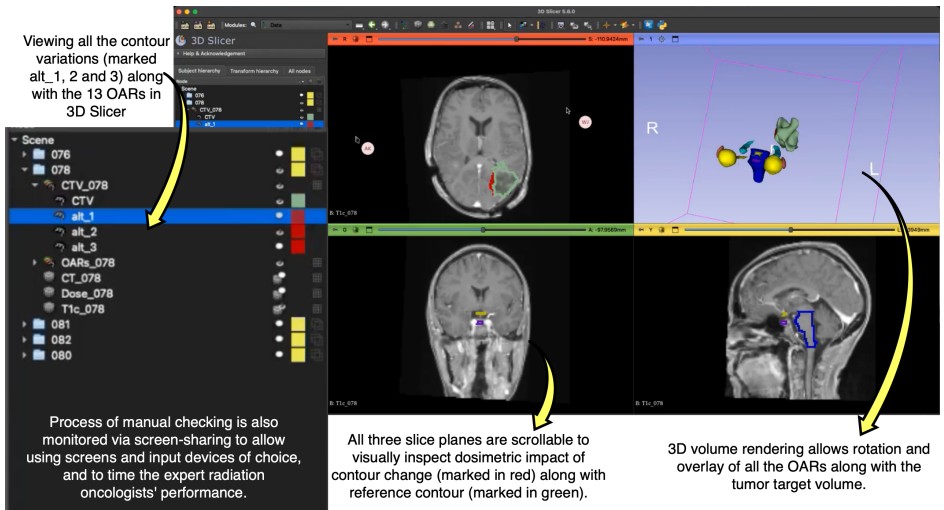

Figure 6: User interface for radiation oncologists to review and classify contour changes (selecting variants via left panel) with three slice plane views and 3D volume rendering. 3D Slicer version 5.6.0.

