# OpenReview forum: "Comparing the Performance of Radiation Oncologists versus a Deep Learning Dose Predictor to Estimate Dosimetric Impact of Segmentation Variations for Radiotherapy"
_MIDL.io/2024/Conference — MIDL 2024 Oral_

### Official Review · Reviewer_6T4f · 2024-02-25

**Confidence:** 4
**Preliminary Rating:** 2
**Final Rating:** 4

**Summary:**

Since over- and under-segmentation of tumor volumes may lead to radiotherapy plans that do not treat the entire tumor and/or exposes certain risk structures to high radiation doses, properly assessing the quality of (deep learning based) segmentation tools is extremely important. The authors of this work investigate the use of an existing deep learning based dose predictor as a new quality metric for such automatic segmentation tools within the field of radiotherapy of brain tumors. Specifically, for the dose prediction model, the authors use 60 patients to train a cascaded UNet mapping the CT volume and the associated segmentations of both organs at risk (OAR) and the tumor to the dose map computed within the software Eclipse (Varian Medical Systems).

Using 54 pairs of tumor segmentation and a segmentation variation, the DL-based dose predictor is compared against expert radiation oncologists to classify whether the shape variation is dosimetrically impactful, i.e., whether the shape variation leads to a positive, negative, or no impact. The experiment demonstrates that the DL-based dose predictor outperforms the three expert radiation oncologists both in terms of accuracy and run-time.

**Strengths:**

Considering that automatic segmentation tools are predominantely evaluated based on geometric metrics, the contribution of the authors is timely and valuable to the community. In addition, the presented method has several advantages:
- It is significantly faster than running full dose simulations and faster than the experts
- The heuristic to classify the output of the dose predictor model is intuitive, easy to implement, and can be calibrated to meet an expected precision / recall
- The authors intend to share the code publicly

**Weaknesses:**

Even though the paper is interesting to read and the topic of quality assessment of automatic segmentation tools is of particular importance, there are several weaknesses to be addressed in the rebuttal:

First of all, the technological novelty may be limited since it seems like the authors presented the same cascaded UNet trained on the same data in a preceding publication. If there are any additional technological contributions, please clarify it.

In addition, the methodology and experiments section lacks details and clarity to replicate and fully assess the paper:
- How were the segmentation variations generated? Are the variations primarily additions to the reference segmentations or are there also subtractions?
- On what basis was the dose threshold for the classification of the impact chosen? Is a 10% change of the maximum dose clincally relevant? How would the results of the experiments change with a different threshold for the ground truth labels?
- How are the ground truth labels generated? If they follow Algorithm 1, what values for alpha and nOAR were chosen? If another algorithm was chosen, please add a short description and justify the difference in algorithms used.
- Figure 4: Over which dataset were the precision and recall values computed?
- Figure 5: What does the yellow error in the right case denote? In general, it is really difficult to understand the images. One easy improvement may be a separate image showing the OARs in the same volume slice and, ideally, a table with the maximum dose values in each OAR.
- Considering the low number of patients, running a cross-validation over the training/validation split could help strengthen the results

Moreover, it is questionable to me whether the comparison between the proposed method and the experts is fair. First of all, the ground truth labels and the DL predictions are generated from dose values. The experts, however, are only seeing the segmentation masks and thus must judge based on geometric proximity, which the authors even state at the beginning of the discussion. The results could be strengthened if, on a subset of the data, the experts are presented with the dose maps.
Second, it is not clear to me whether the 10% change of the maximum dose as a threshold to form the ground truth is clinically meaningful. Considering that the experts are more often stating "no impact" than "sub-optimal" it may also hint that they could accept a higher tolerance. In addition to showing the experts the dose maps, it may be worth to analyse how close to the threshold the ground truth dose values were on the misclassified cases.
Ultimately, the proposed classification method has tuneable parameters nOAR and alpha, which are adjusting the sensitivity of the classifier and could significantly improve the precision and recall. In particular, I am having concerns about alpha being chosen as 0.1, which sets the classification of the impact for a specific OAR to 1% as opposed to 10% for the ground truth generation. Could the authors please comment on this? In addition, I strongly believe that it would strengthen the comparison if error metrics in estimating the dose map for the test set were presented.

Finally, the paper could greatly benefit from a deeper and more critical discussion of the presented results. Please use the above weaknesses as inspiration.

**Detailed Comments:**

- Page 2, second to last paragraph: a deep learning base dose predictor -> a deep learning based
- Fig. 1.: Please replace the graphic with a higher resolved version (> 300dpi)
- Intro to Section 2 & paragraph "contour modifications": In the intro it says that four variations per reference segmentations were produced, while in the paragraph it says "three to four". Could the authors please also explain why the number of variations is varying and why only up to four variations are considered? In theory, unlimited variations could be generated.
- Fig. 3: Please properly label the x and y axis, i.e., prediction (x) vs. target (y).

**Justification Of Final Rating:**

The authors have adequately addressed all of my questions and concerns. I also appreciate the changes to the manuscript, which has noticeably improved the quality of the paper. Even though there is still room for improvement (e.g., the discussion of the results), I am happy to change my initial recommendation to (weak) acceptance.

**Justification Of The Preliminary Rating:**

While I strongly believe that the presented method and study has merit, there are too many details missing to properly assess this work. In addition, the comparison with experts, which is at the core of this paper, seems not to be fair.

**Questions To Address In The Rebuttal:**

Please see the weaknesses section. While I understand that addressing all topics may not be realizable in the short amount of time, I would appreciate if priority is given to adding the missing details of the methodology and providing more insights into the performance of dose predictor and the comparison to the experts.

---

> ### Author Response · Authors · 2024-03-17
> **Author responses to feedback (point-by-point, part 1)**
>
> We thank you for the positive feedback regarding the relevance and potential impact of our work.  Beyond making changes to the text in the revision, here are responses to your comments:
>
> > First of all, the technological novelty may be limited since it seems like the authors presented the same cascaded UNet trained on the same data in a preceding publication. If there are any additional technological contributions, please clarify it.
>
> We would like to clarify that the innovation of the study does not lie within the development of a new dose predictor model, but rather on a novel dosimetry-driven quality control framework. The ability of the model to outperform human experts, using current model architecture, indicates a promising baseline on which to build on improvements. As the framework is the main core message of the study, we refrain from employing more complex architectures/models at this time.
>
> We have clarified and enriched the discussion accordingly.
>
> > In addition, the methodology and experiments section lacks details and clarity to replicate and fully assess the paper:
>
> > How were the segmentation variations generated? Are the variations primarily additions to the reference segmentations or are there also subtractions?
>
> Thank you for the question. We clarify with the text now saying: “For 14 test subjects, between three to four variations to the tumor target volume are made independently by an expert using the same rationale as in (Poel et al., 2021). The reference segmentation follows current delineation guidelines and .....” (last paragraph on page 4). We clarify here further that the variations are primarily additive.
>
> > On what basis was the dose threshold for the classification of the impact chosen? Is a 10% change of the maximum dose clinically relevant? How would the results of the experiments change with a different threshold for the ground truth labels?
> > it is not clear to me whether the 10% change of the maximum dose as a threshold to form the ground truth is clinically meaningful. Considering that the experts are more often stating "no impact" than "sub-optimal" it may also hint that they could accept a higher tolerance.
>
> We thank you for your question. We show below the precision, recall and F1 scores for all three radio-oncologists and our dose predictor model across six different percentage thresholds (20%, 15%, 12%, 10%, 8%, 5%) to create the ground truth:
>
> |                 |                      R1                 |                      R2                 |                      R3                 |                  Prediction             |
> |-----------------|:---------------------------------------:|:---------------------------------------:|:---------------------------------------:|:---------------------------------------:|
> |      Precision  |     0.6, 0.51, 0.45, 0.41, 0.38, 0.78   |     0.61, 0.49, 0.53, 0.48, 0.52, 0.79  |     0.73, 0.56, 0.55, 0.55, 0.54, 0.76  |     0.64, 0.48, 0.57, 0.57, 0.60, 0.78  |
> |      Recall     |     0.65, 0.56, 0.44, 0.35, 0.26, 0.19  |     0.56, 0.50, 0.54, 0.46, 0.43, 0.39  |     0.59, 0.54, 0.57, 0.57, 0.52, 0.50  |     0.56, 0.44, 0.57, 0.57, 0.59, 0.72  |
> |      F1         |     0.62, 0.51, 0.38, 0.29, 0.21, 0.28  |     0.57, 0.50, 0.52, 0.44, 0.42, 0.51  |     0.61, 0.54, 0.56, 0.56, 0.51, 0.59  |     0.58, 0.44, 0.57, 0.57, 0.59, 0.75  |
>
> The Precision of R1 varies between 0.38 and 0.78, R2 between 0.48 and 0.79, R3 between 0.54 and 0.76, and the model between 0.48 and 0.78, with the model being the best at 3/6 thresholds.
>
> For recall and F1-score, the model is best at 4/6 thresholds, indicating that this choice does not change our inference significantly, especially given that there is no current clinical understanding of what constitutes unacceptable levels of dose change in relation to OAR tolerance. We believe for this reason that a 10% change is a reasonable choice, and that in the future,  a dose constraint related metric could help. There are many distinct guidelines (maximum dose, mean dose, volume coverage and so on) which we will incorporate into this work in the next several months.
>
>
>
> > How are the ground truth labels generated? If they follow Algorithm 1, what values for alpha and nOAR were chosen? If another algorithm was chosen, please add a short description and justify the difference in algorithms used.
>
> We apologize for the lack of clarity on this point. The GT labels were produced using Algorithm 1 (alpha = 1, nOAR=1), with the exception that the true dose distributions (from Eclipse) were used instead of those provided by the dose predictor. The predicted labels are generated with alpha=0.1, nOAR=3, with the dose predictor. The parameters are changed for higher robustness: Alpha is set to 0.1 to make the labels more sensitive to overall dose changes, where nOAR is set to 3 to avoid outliers due to a single OAR exceeding the limit.

---

> > ### Author Response · Authors · 2024-03-17
> > **Author responses to feedback (point-by-point, part 2)**
> >
> > > Figure 4: Over which dataset were the precision and recall values computed?
> >
> > These were calculated on the test set.
> >
> >
> > > Figure 5: What does the yellow error in the right case denote? In general, it is really difficult to understand the images. One easy improvement may be a separate image showing the OARs in the same volume slice and, ideally, a table with the maximum dose values in each OAR.
> >
> > Thank you for the comment. This is clarified in the caption now to say “The yellow arrow shows the beam artifacts in the reference dose plan which are not replicated by our model”
> >
> > > Considering the low number of patients, running a cross-validation over the training/validation split could help strengthen the results. Moreover, it is questionable to me whether the comparison between the proposed method and the experts is fair. First of all, the ground truth labels and the DL predictions are generated from dose values. The experts, however, are only seeing the segmentation masks and thus must judge based on geometric proximity, which the authors even state at the beginning of the discussion. The results could be strengthened if, on a subset of the data, the experts are presented with the dose maps.
> >
> > We acknowledge the concerns and clarify that in this first phase we are mostly interested in assessing the current dosimetric awareness of clinicians (i.e., where no dosimetric information is available to them) vs. that of a DL-based approach. We clarify that, according to the clinicians, proximity is not the only “feature” they use in their internal thought process, but rather a combination of experience-based and such proxy features. At this point our main objective is to establish a human baseline that reflects the current clinical scenario. On a second stage, we are indeed conducting what the reviewer refers to.
> >
> > We have revised the discussion section by clarifying the strategy for this and our next steps.

---

### Official Review · Reviewer_rbjR · 2024-02-28

**Confidence:** 4
**Preliminary Rating:** 2
**Final Rating:** 3.5

**Summary:**

The authors propose to automatically predict the impact of tumor segmentation variations on the planned RT dose. This allows them to compare deep learning segmentation and expert segmentation in terms of dosimetric impact rather than the usual segmentation metrics.
The analysis is performed on a small set of CT images (14) from patients with glioblastoma, with multiple modifications of tumor volumes.
Variations on the max dose recorded in the OARs are used to classify the doses of modified volumes as “Sub-optimal”, “No Impact” or “Improved”.

**Strengths:**

The idea/approach is important: evaluating segmentation performance in a more clinically relevant manner than standard segmentation metrics.
Considering OARs for such evaluation is appropriate, as well as comparing with multiple experts.

**Weaknesses:**

1) The method and setup are not very complex, but I found it very hard to understand from Figure 1 and Section 2.2.

2) There is already a validated automatic treatment planning system that you use to create the three GT categories "Sub-optimal", "No impact", "Improved". I don't understand the motivation for the analysis: creating a new DL dose predictor and comparing the OAR dose impact with this GT (and with the human dose impact prediction).

3) How can you say that just lower OAR dose is an improved dosimetry (and vice-versa), without considering the dose in the tumor?
With this definition, a smaller volume results in an "improved" dose, and no volume at all is a perfect dose ?

4) 54 contour variations are derived from 14 test volumes.
How are the modifications made? I think it is a very important point for the study and it is not mentioned.

5) The test set is very small (derived from 14 volumes).

6) There are incoherences (e.g. number of derived volumes from the test cases, see below).

**Detailed Comments:**

"which is time-consuming when done manually. Fully manual segmentation is time-consuming and can take up to seven hours" repeated.

DSC stands for Dice Similarity Coeffiient. Also, throughout the document, use DSC instead of Dice.

"16 cases for future evaluation" Explain why.

Between three to four variations to the tumor target. Why does this vary? Somewhere else in the paper, you say 4 (Section 2)

"10% percent"

"Table. 3 shows the weighted average" should be Table 1.

"inter-rater variability in performance" Actual inter-rater variability would be more relevant.

**Justification Of Final Rating:**

The authors replied to my questions and clarified some of my concerns. The overall quality of the manuscript is improved. Some limitations remain, but I changed my recommendation to borderline accept.

**Justification Of The Preliminary Rating:**

The idea of clinically relevant segmentation evaluation is important, but the paper does not seem ready for publication.
Many clarifications, a better and more systematic evaluation (more cases, less subjective and better motivated classification of dose impact, description of the volume modifications), and maybe a change of evaluation are required in my opinion.

**Questions To Address In The Rebuttal:**

All of the main concerns in the "Weaknesses" part.
I may have missed an important point that would answer my second point, in which case a clarification is welcome.

---

> ### Author Response · Authors · 2024-03-17
> **Author responses to feedback (point-by-point, part 1)**
>
> We thank you for the detailed feedback and suggestions, allowing us to improve the manuscript. Here are our responses to points you raise:
>
> > The method and setup are not very complex, but I found it very hard to understand from Figure 1 and Section 2.2.
>
> > There is already a validated automatic treatment planning system that you use to create the three GT categories "Sub-optimal", "No impact", "Improved". I don't understand the motivation for the analysis ...
>
> We appreciate your feedback and apologize for the lack of clarity. Our work is motivated by the evolving role of radiation oncologists in the era of AI-assisted radiotherapy. As automated segmentation becomes more prevalent, the role of medical experts is shifting from manual segmentation to monitoring and correcting AI-generated contours.
>
> In this new role, it is crucial for experts to assess if such contours’ deviation from a hypothetic ground truth may affect the dose delivered to tumor or organs at risk in a clinically-relevant manner, i.e. altering the chances of tumor control or the risk of normal tissue toxicity.. In order to make such a comparison, one would have to prepare a treatment plan based on the AI-generated contours and assess the dose delivered to the ground truth organs and tumor in this plan, and compare it to the achieved doses when planning directly to the ground truth structures. However, due to the time-consuming nature of treatment planning and common lack of an immediately available ground truth structure to compare AI-generated contours with, such comparisons are not feasible in routine clinical practice.  Our proposed deep learning-based dose predictor addresses this challenge. It enables a novel paradigm where alternative segmentations can be ranked from a dosimetric perspective, making the comparative assessment more efficient and feasible.
>
> We have adapted the introduction to make the motivation of our work clearer.
>
> > How can you say that just lower OAR dose is an improved dosimetry (and vice-versa), without considering the dose in the tumor? With this definition, a smaller volume results in an "improved" dose, and no volume at all is a perfect dose ?
>
> The ground truth plans are constructed with the target volume receiving the full prescribed dose, normalized for the same coverage. This means that no target coverage is sacrificed when OAR dose is estimated. The metric for the deep learning-based dose predictor hence is subject to this criterion, which will fail if it predicts no target volume dose, thereby marking everything “improved”.
>
> We have revised the text accordingly to clarify this.
>
> > 54 contour variations are derived from 14 test volumes. How are the modifications made? I think it is a very important point for the study and it is not mentioned.
>
> We fully agree and acknowledge the concern. We have revised the text to clarify that the modifications were made manually by an expert, following a similar rationale as presented in Poel et al. 2022 and Poel et al. 2021, where variations follow clinically plausible variations.
>
> Poel R., et al., Impact of random outliers in auto-segmented targets on radiotherapy treatment plans for glioblastoma. Radiation Oncology, 17, 1-18, 2022.
>
> Poel R., et al., The predictive value of segmentation metrics on dosimetry in organs at risk of the brain. Medical Image Analysis, 73, 102161, 2021.
>
> > The test set is very small (derived from 14 volumes).
> We agree with this assessment. We, however, note that scheduling clinician's time for running these human-versus automated-model evaluations is not easily scalable. The segmentation and planning preparation process consumes more than 70% of the total time from diagnosis to treatment, typically spanning over two weeks. Consequently, producing meaningful alternate segmentations and re-planning doses using the clinical TPS is a significant time investment. We are aware of this limitation and are working towards enriching the dataset of cases, and believe this first evidence is of high interest to the scientific community.
>
> We note that the work of Poel et al. MedIA 2021, employed a total of 12 cases from which a similar multiplicative effect exists leading to a total of 60 alternatives RT struct sets (our total set comprises 54 RT struct sets). We also highlight the multi-expert evaluation presented in this paper, which in our opinion favorably compares to other multi-rater studies presented in our community and similar venues.
> Poel R. et al., The predictive value of segmentation metrics on dosimetry in organs at risk of the brain. Medical Image Analysis, 73, 102161, 2021.

---

> > ### Author Response · Authors · 2024-03-17
> > **Author responses to feedback (point-by-point, part 2)**
> >
> > > There are incoherences (e.g. number of derived volumes from the test cases, see below).
> > > Between three to four variations to the tumor target. Why does this vary? Somewhere else in the paper, you say 4 (Section 2)
> >
> > Thank you for the question and apologies for the lack of clarity.  We have two test cases with only three (instead of four) alternatives because the tumor target volume was small, and making more alternatives would not be feasible, as they would not be distinct enough.
> >
> > > "inter-rater variability in performance" Actual inter-rater variability would be more relevant.
> >
> > Inter-rater variability amongst the three experts show the correlation is between weak and moderately positive (based on spearman rank coefficient): R1 and R2: statistic=0.4779, pvalue=0.0002; R1 and R3: statistic=0.3304, pvalue=0.0146; R2 and R3: statistic=0.6657, pvalue=3.9449e-08; This shows variability while also general agreement on the categories. (0.0 to 0.3 is considered weak, and 0.4 to 0.7 is considered moderate correlation).
> >
> > > "16 cases for future evaluation" Explain why.
> >
> > We are planning to use these cases for a future evaluation where we show dose maps (like suggested by reviewer 3) and additional heatmaps (specifically related to ASTRA, Kamath et al. 2023) along with the contour variations and evaluate if the human experts perform better than earlier, and, if there is higher inter-rater agreements between them than earlier.

---

### Official Review · Reviewer_P1Jg · 2024-03-03

**Confidence:** 4
**Preliminary Rating:** 4
**Recommendation:** Poster
**Final Rating:** 4

**Summary:**

In this interesting manuscript, the authors describe a deep learning model that estimates the dosimetric impact of segmentation variations in radiotherapy. The problem and the rationale of the work is well explained and the paper is well written. The results compare the model to three expert radiation oncologists and show promising results. Great work!

**Strengths:**

- The work is well motivated and contains references to many previous studies on this topic
- Clinically relevant problem
- Novel approach to this problem
- Comparison to three expert radiation oncologists
- Code will be made available

**Weaknesses:**

- The establishment of the ground truth segmentations is unclear to me.
- The test set actually only consists of 14 cases, which is relatively small.
- An external validation on data from another center would strenghten the paper.

**Detailed Comments:**

- Finally, there are 54 variations on 14 test cases, because the authors write that 3 to 4 variations are performed. Why not make it consistent and apply four variations to all test cases, so that 56 variations are obtained?
- Please clarify how the reference segmentations are set. Since the authors also write that there is substantial interreader variability (Results section reads "Notably, we underline the high inter rater variability in performance among the three experts."), it is important to make it very clear how the reference segmentations are obtained.
- I find the naming of the three categories (“Sub-optimal”, “No Impact” or “Improved”) suboptimal. I would suggest to rename to something more clear and simple, perhaps just "Worse, no change, Better", or "Detoriated, no change, Improved".

**Justification Of Final Rating:**

I would like to thank the authors for their responses and the rebuttal. My questions have been adequately answered! The paper would benefit from a larger scale validation, but I understand that that takes time and effort.

**Justification Of The Preliminary Rating:**

This work is well motivated and solid scientific work. The developed deep learning model that estimates the dosimetric impact of segmentation variations in radiotherapy is an interesting approach. I think this paper will be of interest to the MIDL community.

**Questions To Address In The Rebuttal:**

Please address my points in the detailed comments.

**Special Issue:**

No

---

> ### Author Response · Authors · 2024-03-17
> **Author responses to feedback (point-by-point)**
>
> We thank the reviewer for the positive feedback on our work and hope our revision based on these suggestions can lead to further improvement in the review score. Here is a summary of our responses:
>
> > The establishment of the ground truth segmentations is unclear to me.
>
> Thank you for this remark. We clarify with the text now saying: “The reference segmentation follows the current delineation standards.” (last paragraph of page 4).
>
> > The test set actually only consists of 14 cases, which is relatively small.
>
> We agree with this assessment. We, however, note that scheduling clinician's time for running these human-versus automated-model evaluations is not easily scalable. The segmentation and planning preparation process consumes more than 70% of the total time from diagnosis to treatment, typically spanning over two weeks. Consequently, producing meaningful alternate segmentations and re-planning doses using the clinical TPS is a significant time investment. We are aware of this limitation and are working towards enriching the dataset of cases, and believe this first evidence is hypothesis generating and of high interest to the scientific community.
>
> We note that the work of Poel et al. MedIA 2021, employed a total of 12 cases from which a similar multiplicative effect exists leading to a total of 60 alternatives RT struct sets (our total set comprises 54 RT struct sets). We also highlight the multi-expert evaluation presented in this paper, which in our opinion favorably compares to other multi-rater studies presented in our community and similar venues.
> Poel R., Rufenacht E., Herrmann E., Scheib S., Manser P., Aebersold D., and Reyes M. The predictive value of segmentation metrics on dosimetry in organs at risk of the brain. Medical Image Analysis, 73, 102161, 2021.
>
> > An external validation on data from another center would strengthen the paper.
>
> Thank you for the suggestion. We do have plans to run this analysis on external data, and since scheduling clinician time to arrange these subjective evaluations is not easy, we will run more of these in the weeks and months to come.
>
> > Finally, there are 54 variations on 14 test cases, because the authors write that 3 to 4 variations are performed. Why not make it consistent and apply four variations to all test cases, so that 56 variations are obtained?
>
> Thank you for the question and apologies for the lack of clarity.  We have two test cases with only three (instead of four) alternatives because the tumor target volume was small, and making more alternatives would not be feasible, as they would not be distinct enough.
>
> > Please clarify how the reference segmentations are set. Since the authors also write that there is substantial interreader variability (Results section reads "Notably, we underline the high inter-rater variability in performance among the three experts."), it is important to make it very clear how the reference segmentations are obtained.
>
> Thank you for the recommendation. We clarify with the text now saying: “The reference segmentation follows the current delineation standards.” (last paragraph of page 4). For inter-reader variability amongst the three experts, the correlation is between weak and moderately positive (based on spearman rank coefficient): R1 and R2: statistic=0.4779, pvalue=0.0002; R1 and R3: statistic=0.3304, pvalue=0.0146; R2 and R3: statistic=0.6657, pvalue=3.9449e-08; indicating that there is variability, but they all generally agree on the categories. (0.0 to 0.3 is considered weak, and 0.4 to 0.7 is considered moderate correlation), supporting our statements.
>
> > I find the naming of the three categories (“Sub-optimal”, “No Impact” or “Improved”) suboptimal. I would suggest to rename to something more clear and simple, perhaps just "Worse, no change, Better", or "Detoriated, no change, Improved".
>
> Many thanks for this suggestion. We have renamed the categories: “Worse, No Change, Better”.

---

### Meta-Review · Area_Chair_5JZA · 2024-04-03

**Recommendation:** Accept (Poster)
**Confidence:** 5

**Metareview:**

The authors describe a deep learning model that estimates the dosimetric impact of segmentation variations in radiotherapy. The problem and the rationale of the work is well explained and the paper is well written. During the rebuttal phase, the authors adequately addressed many of the concerns raised, which helped in improving the quality of the paper.

---

### Decision · Program_Chairs · 2024-04-05

Accept (Oral)